# Mesenchymal Cell Growth and Differentiation on a New Biocomposite Material: A Promising Model for Regeneration Therapy

**DOI:** 10.3390/biom10030458

**Published:** 2020-03-16

**Authors:** Leslie Pomeraniec, Dafna Benayahu

**Affiliations:** Department of Cell and Developmental Biology, Sackler School of Medicine, Tel Aviv University, Tel Aviv 69978, Israel; lesliepo@mail.tau.ac.il

**Keywords:** mesenchymal cells, scaffold, biocomposite, tissue regeneration

## Abstract

Mesenchymal stem cells serve as the body’s reservoir for healing and tissue regeneration. In cases of severe tissue trauma where there is also a need for tissue organization, a scaffold may be of use to support the cells in the damaged tissue. Such a scaffold should be composed of a material that can biomimic the mechanical and biological properties of the target tissues in order to support autologous cell-adhesion, their proliferation, and differentiation. In this study, we developed and assayed a new biocomposite made of unique collagen fibers and alginate hydrogel that was assessed for the ability to support mesenchymal cell-proliferation and differentiation. Analysis over 11 weeks in vitro demonstrated that the scaffold was biocompatible and supports the cells viability and differentiation to produce tissue-like structures or become adipocyte under differentiation medium. When the biocomposite was enriched with nano particles (NPs), mesenchymal cells grew well after uptake of fluorescein isothiocyanate (FITC) labeled NPs, maintained their viability, migrated through the biocomposite, reached, and adhered to the tissue culture dish. These promising findings revealed that the scaffold supports the growth and differentiation of mesenchymal cells that demonstrate their full physiological function with no sign of material toxicity. The cells’ functionality performance indicates and suggests that the scaffold is suitable to be developed as a new medical device that has the potential to support regeneration and the production of functional tissue.

## 1. Introduction

Tissue regeneration relies on the differentiation of stem cells, however, tissue does not always spontaneously recover after trauma. In such cases, there is a need to aid tissue-repair by introducing a scaffold to promote cell growth and differentiation. An implanted scaffold should support autologous cells to progressively enter the scaffold and to form an extracellular matrix (ECM) that will eventually replace the scaffold and integrate with the regenerated tissue.

Tissue-engineered scaffolds are designed to mimic the structure of organs and enable controlling the physical properties of the scaffold material. The aim of this approach is to avoid the use of autologous grafts, which are limited by the availability of the patient’s own tissue and avoid an additional surgical procedures. Since synthetic polymers do not integrate with the body and may trigger an immune response and chronic inflammation [1], the use of biocompatible natural materials as a scaffold material is considered beneficial.

Natural materials are similar to biological macromolecules, which minimize immunological reactions and chronic inflammation [2,3,4]. The biopolymers provide 3D structures that facilitate the stem cell attachment, allowing their proliferation and consequently promotes the formation of new tissue to improves regeneration and healing processes [1,5,6,7,8,9,10,11]. The use of naturally made scaffolds is the best approach, but they may lack the physical/mechanical properties needed for the new tissue formed.

We have developed a composite biomaterial scaffold made from collagen fibers embedded in alginate hydrogel [12,13,14,15,16,17]. The collagen used as a biopolymer for producing implantable medical devices is usually obtained through chemical extraction and reconstitution processes, resulting in products with poor mechanical strength [18,19,20]. However, the biocomposite we describe here focuses on the unique coral-derived collagen fibers that were recently identified for their molecular composition and structure [13,14] and mechanical properties [12,15,16,17]. These attributes make our biocomposite uniquely suitable for use in a new generation of scaffolds that can be tailored to meet the mechanical properties of various target tissues. The biological and mechanical properties of these collagen fibers facilitate the desired physiological activity, together with the promotion of cell proliferation and differentiation. The collagen fibers are embedded in an alginate hydrogel, a polysaccharide extracted from algae that has already been applied in a wide spectrum of uses such as food, pharmaceutics, and medical device industries [21,22,23,24]. The addition of the collagen fibers reinforces the alginate in the bio composite and its advantageous properties facilitate cell attachment to provide temporary bio functional support while regeneration of tissue takes place.

To date, a number of marine-derived biopolymers have been proven to be safe for clinical use [25,26,27,28]. In this context, we have also developed a production protocol that enables us to include also nanoparticles (NPs) in order to enrich the scaffold with growth factors or a potential drug. Such a biocomposite scaffold should mimic the natural structure of the tissue that would replace and also may provide temporary biofunctional support for the cells.

The in-vitro models described here allow us to determine the conditions required for mesenchymal cell growth and to monitor cellular viability, proliferation, and differentiation required to regenerate a functional-tissue. The cells and the material interface are dynamic, since cells sense the material via adhesion molecules and react to the material’s stiffness and nanostructuring. The material was tested for biocompatibility in vitro to meet the determined ISO-10993 multi-standard, established by the Food and Drug Administration Organization. In this study, we describe a variety of in-vitro assays that have proven the pivotal physiological function of the mesenchymal cells in the biocomposite to become a functional tissue.

## 2. Material and Methods

### 2.1. Cell Culture

Mouse mesenchymal 3T3-L1 cells were cultured in a growth medium consisting of Dulbecco’s modified Eagle’s medium (450 mg/dL), 10% fetal bovine serum, 1% l-glutamine, 0.1% penicillin-streptomycin, and 0.5% 4-(2-hydroxyethyl)-1-piperazine-ethanesulfonic acid (HEPES; Biological Industries, Beit Haemek, Israel)

### 2.2. Cell Differentiation to Adipocytes

Differentiation was induced by incubating cells in growth medium supplemented with 100 U/mL of insulin (Biological Industries), 1 μM dexamethasone (Sigma-Aldrich) and 400 μM 3-isobutyl-1-methylxanthine (Sigma-Aldrich). After two days, the medium was replaced with growth media supplemented with 100 U/mL insulin. The adipocyte differentiation phenotype is based on cell morphology and lipid droplets accumulation as was previously detailed and described by Mor-Yossed Moldovan et al. [29].

### 2.3. Nanoparticles (NPs)

Fluorescein isothiocyanate (FITC)-labeled silica (Si) NPs of a 250 nm size were previously studied for their effect on mesenchymal cells [30]. Here we used the FITC-Si-NPs in two assays: (1) Mesenchymal cells were cultured in a 2D system and incubated with Si-NPs for 3 h. (2) The FITC-Si-NP were added to the bio-composite and incubated for 24 h. After incubation times in the different system models the medium was removed and replaced with fresh growth medium for further cell growth and monitoring by live imaging for several weeks.

### 2.4. Collagen Fibers

Collagen fibers were isolated by mechanical extraction from *Sarchophyton* soft coral [12,13,14]. The fibers were extracted from the soft coral colonies that were torn to expose the fibers. The fibers were then physically pulled out from the soft coral [12] and the isolated fibers were manually spun around a thin polylactic acid (PLA) frame, to create a dense net of multidirectional fiber bundles. The extracted fibers were washed thoroughly in a series of solutions [distilled water, 0.1% sodium dodecyl sulfate, 0.5 M ethylene di-amine-tetra acetic acid (EDTA), and phosphate buffered saline (PBS)], and then immersed in ethanol 70%.

### 2.5. Alginate/Collagen Biocomposite Fabrication

By dissolving sodium alginate (Protanal LF 10/60, FMC Biopolymer, Philadelphia, PA, USA) in double distilled water, 1.5% or 3% (*w*/*v*) alginate solution was produced. For some experiments, Si-NPs or cells were mixed in the alginate (Figure 1A). When combining cells in the biocomposite, they were added to the mixture just before it was used to immerse the collagen fibers (Figure 1B). The isolated fibers on the frame were embedded in alginate (Figure 1C,D), and then inserted into a dialysis membrane (6000–8000 MWCO, Spectra Por, Spectrum Labs Inc., Rancho Dominguez, CA, USA). The membrane was sealed, flattened, and soaked for 24 h in a solution containing calcium at physiological concentration (0.02 M CaCl_2_ or growth medium for cell enriched scaffolds; Figure 1E–G). Calcium divalent cations mediate cross-linking between the polysaccharide chains of the alginate, which become a hydrogel. The bio-composite was removed from the membrane and the frame (Figure 1H). To analyze the bio-composite stability, the material was air dried, or immersed in 70% ethanol or growth media.

### 2.6. Microscopy and Imaging Analysis

(1) Live cell cultures and biocomposites were observed under phase contrast microscopy and photographed digitally (Optiphot, Nikon, Tokyo, Japan), or under EVOS FL Auto 2 Imaging system (Thermo Fisher Scientific, Walthman, MA, USA). (2) Scanning electron microscopy (SEM) imaging was performed as previously described [31]. Briefly, samples were fixed in 2.5% glutaraldehyde and dehydrated in increasing concentrations of ethanol (30–100% for 10 min each) then air dried at RT for 30 min. For observation samples were mounted on aluminum stubs, coated with gold in a sputtering device for 3 min at 15 mA and analyzed under HR-SEM (Jeol JSM 6700, Tokyo, Japan).

### 2.7. Analysis of Cell Growth Directionality and Orientation Mapping

Cellular arrangement orientation and directionality was evaluated from digital pictures by ImageJ software (https://imagej.net/Directionality, by Jean-Yves Tinevez, V2.0, NIH, Bethesda, MD). Phase images were converted to binary and various ranges of interest (ROIs) were selected. ROIs were processed with a fast Fourier transform (FFT) filter and bright pixels were analyzed with a directionality analysis function.

## 3. Results

Scaffolds are designed to serve as a temporary structure for cells to promote tissue regeneration at injured sites and to support cells during the repair process. The scaffold provides a microenvironment suitable for cell proliferation and differentiation.

The biocomposite of collagen and alginate produced as described in the Material and Methods section (Figure 1) was assessed for biocompatibility in vitro and for its ability to support mesenchymal cells proliferation, growth, and differentiation. Mesenchymal cell morphology and growth were monitored in response to the substrate’s stiffness, thus applied mechanical forces on the cells leads to cytoskeleton reorganization. The mechanical properties of the biocomposite matrix affect the cell activity and shape, the alginate hydrogel provides a low stiffness and a 3D microenvironment niche. For cells mixed in the alginate, equal forces were applied all over the cell surface, resulting in non-directional and low stress fiber arrangement and affect the cells to a spherical shape (Figure 2A,B). In contrast, the adhesion of the cells to the collagen fibers that were embedded in the hydrogel of the biocomposite or to neighboring cells produced a stiffer substrate. When the cells interacted with a stiffer substrate, it resulted in cellular cytoskeleton reorganization and their orientation form the stress fibers lead the cells to become elongated featuring a fibroblast-like morphology. Thus, cells growing on the biocomposite interacted with the altered niches that proposed combined alginate and collagen fibers, and reacted respectively: cells in contact with collagen fibers became elongated, whereas those growing in low-stiffness alginate observed with a circular morphology (Figure 2C). In this way, the alginate concentration and the density and organization of the collagen fibers in the bio composite supplied the microenvironment for the cells and affect their shape and consequently, their differentiation. The consistency of the composite material had a pivotal influence on cell morphology and function depending on the density and organization of the collagen fibers and based on the alginate concentration. When the alginate became a 3D hydrogel in the presence of Ca ions, it still had pores in the hydrogel that enable cell motility and the diffusion of nutrients while the collagen fibers reinforced the hydrogel structure. The cells were motile and proliferated within the biocomposite scaffold and were monitored in-vitro. Mesenchymal cells were analyzed for their invasion into the hydrogel, as well as their proliferation and performance over time when cultured in biocomposite of collagen fibers embedded in 1.5% or 3% alginate hydrogels. We took two approaches, in the first one the cells were mixed with the alginate to become embedded in the biocomposite during hydrogel gelation and the second, where the cells were seeded on a ready-made composite and followed in vitro for several weeks. We found that both concentrations of alginate allowed us to maintain cellular viability, and the cells continued to divide and proliferate giving rise to an accumulation of cells within the biocomposite. Dividing cells were visible in the alginate (Figure 2E,H) and also among those cells that attached to collagen fibers (Figure 2I). The cells clustered in the alginate appeared spherical (Figure 2D) but formed tissue-like structures when growing on or between the fibers (Figure 2E–G). In the experiments where the cells were seeded on top of the biocomposite (after gelation), we could visualize proliferating cells in the 1.5% alginate and tissue-like structures at day seven of the culture, whereas in the 3% alginate, the cell clusters and organized structures between fibers after a prolonged culture time of 2 weeks. After a follow-up period of 11 weeks, tissue-like structures were visible along and between fibers, forming complex layers of cells (Figure 2E–G). We noted that cells migrated through the alginate and reached the rigid surface of the culture plate, indicating that both alginate concentrations provided a beneficial 3D structure suitable for a continuous cell migration and growth support.

With continued culture, the cells could be seen growing preferably on the collagen fibers, populating the fibers and formed tissue-like structures (Figure 2G and Figure 3A). It was clear that the cell mass on the collagen fibers expanded along or to neighboring fibers, thus creating a cellular network. When the growing 3D cellular structures were examined, dividing cells could be visualized in the periphery (Appendix A), while the internal layers displayed stretched cells. The cellular arrangement in different locations on the developing tissue-like structures were evaluated according to the cells’ morphology and directionality. The orientation of the mesenchymal cells growing next to a fiber (Figure 3B) tended to be organized parallel to the fiber featuring a higher distribution in a direction ±80° (Figure 3B). In contrast, cells growing at a distance from the fibers were organized at an angle between crossing fibers to, forming a tissue-like structure reflected as a multidirectional elongation shape on a histogram (Figure 3C). These differences illustrate the complex cellular interactions between cells on the fibers and between neighboring cells.

Medical devices for tissue regeneration may provide biological cues to promote cell proliferation and differentiation. Previously, we demonstrated that the arrangement and orientation of the collagen fibers in the biocomposite could be designed to produce tailor-made scaffolds with the desired mechanical behavior. In addition, the alginate component could be used to entrap nanoparticles (NPs) in order to allow us to enrich such a medical device with molecules, such as tissue-specific growth factors, drug therapy, for the purpose of regulating cellular activity, or differentiation towards a desired lineage fate.

The NPs were introduced into the alginate solution when producing the biocomposite, as presented in Figure 1, but alternatively they could be added to cells already in culture. Here, we used fluorescence imaging to monitor the uptake of FITC-Si-NPs into cells by endocytosis (Figure 4). Cells incubated with FITC-Si-NPs maintained proliferation over several weeks and readhered after release by trypsin and continued to proliferate and differentiate. Cells with FITC-Si-NPs could still be visualized 24 days after a one-time incubation with particles that were introduced into the culture medium and taken up by the cells via endocytosis. FITC-Si-NPs in the cultured mesenchymal cells was visible in the 1.5% alginate biocomposite analyzed after 24 h (Figure 4). Clusters of FITC-Si-NPs were visualized in populated areas (Figure 4A) and could be followed over time as the clusters dispersed. Labeled cells were visible even four weeks after the one time NP treatment. During this time, the mesenchymal cells continued to proliferate with no visible signs of toxicity (Figure 4A,B).

Scaffold materials designed to support cells for biomedical applications needed to be biocompatible with no cytotoxicity. As shown here, the in-vitro model tested the cell viability, proliferation, morphology, and differentiation capability of mesenchymal cells cultured on the biocomposite for up to 11 weeks. Figure 2, Figure 3 and Figure 4 present the results of cell division, proliferation, and formation of tissue-like structures. When the biocomposites were transferred from the culture plate after three weeks of culture, we noted cells migrating through the alginate to the rigid surface of the culture dish and continued to grow (Figure 4E). The migrating cells were well spread and had a flat morphology with a few round cells undergoing division. The cells grew in multilayers and exhibited strong cell-to-cell adhesions that resembled “bridges” in less populated areas (Figure 4E). The cells conserved their proliferation capability, shown by the dividing cells in the culture and eventually reached high confluency. Notably, cells that had endocytosed FITC-Si-NPs were metabolically active during this period, even on the 2D culture plate (Figure 4F).

The mesenchymal cells used in this study were preadipocytes with fibroblastic morphology, elongated shaped which differentiate into adipocytes under differentiation medium. The cells fate transition was reflected by morphological alterations in the cell shape to a round form and became adipocyte that was noted by accumulation of lipid droplets (LDs). This morphological observation could be verified without the need for a specific stain (which was not possible in the alginate biocomposite). Adipogenesis (i.e., lipid droplets accumulation) occurred in the 3D biocomposite cultures (Figure 4C,D) as well as in cells that migrated out from the biocomposite onto the 2D culture dish (Figure 4G,H). Thus, over a few weeks of culture, cells proliferated and differentiated to mature adipocytes and were morphologically recognized both on the 2D culture plate (Figure 4G,H) and in the biocomposite (Figure 4C,D and Appendix A). Adipogenesis progressed up to 11 weeks of culture mimicking physiological tissue formation. The results clearly demonstrated that the new biocomposite composed of collagen fibers and alginate is able to maintain cells in culture for a long period of time of over 3 months. During this period, the mesenchymal cells were physiologically active, viable, proliferated, and retained the capacity to differentiate. These results satisfy the IS0 10993-5 standard for cytotoxicity needed during the production of a medical device.

## 4. Discussion

The biocomposite described here is made of two natural polymers: a unique collagen fibers isolated from a soft coral [12,15,16,17], and alginate. Alginate is a biocompatible polysaccharide composed of (1,4)-linked β-d-mannuronate (M) and α-l-guluronate (G) residues that is widely used in medicine for wound healing, drug delivery, and tissue engineering [21,22,23,24]. The stability of the alginate hydrogel relies on the physiological concentration of calcium divalent cations that mediate alginate crosslinking to become hydrogel. In addition, the calcium diffusion in the physiological microenvironment supports the hydrogel structure and provides the stability of alginate hydrogel demonstrated here over several weeks in in-vitro culture. This is of a great importance when designing scaffolds for tissue regeneration and physiological activity as well as for stability. The scaffolds are also designed to possess specific mechanical properties to stimulate the desired cell growth and their differentiation. Cells react to mechanical forces applied by the microenvironment through mechanoreceptors, which transduce the forces to produce cytoskeleton reorganization. As a consequence, cells in contact with rigid surfaces present different cell adhesion protein in comparison to those growing on a soft surface. Thus, the mechanical characteristics of the biocomposite matrix are critical for cell fate determination. The mechanical profiles of these materials were characterized in an earlier study [12] where the ultimate tensile strength (UTS) of the collagen fibers was shown to be 39–59 MPa, which is three orders of magnitude higher than that of the 3% alginate hydrogel, whereas the parameters for collagen-alginate exhibit hyperelastic behavior similar to that of the collagen fibers alone. Thus, the collagen-alginate biocomposite presents a combined mechanical scenario whereby the collagen fibers provide stiffness to the composite, and the softer alginate serves as a connecting matrix that entraps the fibers and cells. We were also able to design scaffolds in-silico based on the analyzed mechanical properties of the biocomposite, in order to provide defined mechanical characteristics and meet future needs for specific target tissues [16,17,32,33].

The different mechanical characteristics of the materials are well reflected by the cellular morphology of the included cells. As shown in Figure 3, cells growing in a 3D low-stiffness alginate hydrogel, were spherical due to the similar mechanical forces applied all over the cell surface. The 2D highest-stiffness is provided by the surface of the collagen fibers, which were endowed with cell adhesion motifs to promoted better contact with the substrate (the collagen fibers) [12,14]. Cytoskeleton reorganization lead to the formation of stress fibers, which resulted in a more elongated and more dispersed morphology (eccentricity). Cells growing on the interface between the collagen fibers and hydrogel matrix were elongated due to adhesion to the fibers, but were semispherical as a result of the mechanical influence of the alginate. Thus, our results demonstrate that matrix stiffness plays a pivotal role in cell eccentricity, as stress fibers are formed intracellular. In addition to mechanical differences, unlike alginate, the cell adhesion motifs on the collagen fibers promoted stable cell attachment. The cells tend to attach to a fiber instead of growing in the alginate component. This biological preference was exhibited in the biocomposites throughout the cell culture, as demonstrated in Figure 2, where the cell mass found in the alginate decreased after the cell seeding step in favor of developing tissue-like structures between the collagen fibers. Cells supported the proliferation of neighboring cells by secreting growth factors to the microenvironment. Due to the consistency of the alginate matrix, these factors diffused slowly and their concentration is higher in the vicinity of the secreting cells, which promotes growth in clusters. The cell clusters were noted as tissue-like structures when growing on the fibers and as spherical aggregates when growing in the alginate only (Figure 2D–G). The cells also remodeled their microenvironment by secretion of ECM proteins, which facilitated cellular adhesion and also contribute to the observed cluster growth characteristics.

Mesenchymal cells cultured in biocomposites grew well and succeeded in proliferating and migrating in the alginate (Figure 2). Interestingly, the cells displayed a spherical shape when growing in the hydrogel until they reached and spread on the collagen fibers, where they acquired a more elongated shape and could be seen to organize into a tissue-like structure (Figure 2D–G). This phenomenon is suggested to be derived from the differential mechanics between the collagen fibers and the alginate. Attachment of the cells to the fibers causes changes in their morphology and proliferation rate, based on the material mechanical properties that affect the cytoskeleton rearrangements. Cell adhesion to fibers enabled the cytoskeleton to contract and the cells migrate faster along the fibers, instead of dispersing in a multidirectional migration through the alginate. Cells that grew and propagated between fibers were elongated along, and parallel to the fibers, while between fibers they displayed a multi-directional arrangement. This is especially evident for cells growing between crossing fibers, which meet at different directions to create an angular connecting structure on the surface of the fibers and the cells form the tissue like structures (Figure 4A and Appendix A). We could also detect dividing cells and cells in contact with the alginate that presented a semispherical morphology.

In this study, we also described a new concept of adding NPs into the production process of the biocomposites in order to serve as a delivery system, for example by conjugating NPs and growth factors for biological and chemical signals in the scaffold. We demonstrated endocytosis uptake of FITC-NPs by the cells and following our earlier study, we reported that NPs could induce gene activation, proliferation, and differentiation in human mesenchymal cells [29]. We followed mesenchymal cells uptake of the FITC labeled NPs and evaluated their effect on cell viability and morphology in a 3D biocomposite and in 2D in-vitro systems over time. The follow-up over several weeks demonstrate that cells incubated with NPs proliferated and were able to readhere after trypsinization, the cells were splitting and when transfered to new cultures were viable and continue to proliferate well. Cells in the biocomposite that internalized the NPs by endocytosis (Figure 4), continued to proliferate in culture over a period of several weeks. The observation that NPs can be introduced into the 3D hydrogel biocomposite, is of fundamental importance for scaffolds in tissue repair, where the scaffold needs to interact with the native tissue to incorporate cells and physiological molecules like growth factors. Thus, the 3D model of cells incubated with NPs in the alginate demonstrates the great potential of the system in the design of a biomedical device for in-vivo tissue regeneration.

## 5. Conclusions

The advantage of newly developed biocomposite had no cytotoxic effects. Even when the cells were treated with NPs they continued to grow and differentiate in 2D and 3D biocomposite models. The biological cues can be introduced by adding NPs for the delivery of growth factors, to modulate cell proliferation and differentiation into a desired lineage fate. Specifically, we followed the mesenchymal cell differentiation fate into adipocytes as assessed by the morphology and accumulation of lipid droplets. This process was maintained for several weeks in the 3D hydrogel and for even longer times in cells that invaded the hydrogel and migrated towards the culture dish surface. These cells were successfully differentiated into adipocytes at both the culture plate and the 3D biocomposite that was assessed by the adipocyte cells’ shape and the accumulation of lipid droplets. Taken together, the presented collagen-alginate biocomposite was found to be safe for cell proliferation, growth, and differentiation, with no signs of cytotoxicity or stress. These allow us to demonstrate that the combination of collagen fibers and alginate hydrogel produced a biocomposite with the ability to support mesenchymal cell growth and differentiation and meet the criteria of the FDA IS0 10993-5 standards for cytotoxicity and biocompatibility. The mesenchymal cells function well, in this newly developed biocomposite emphasizing its high potential material for the development of biomedical devices for tissue therapy.

## Figures and Tables

**Figure 1 biomolecules-10-00458-f001:**
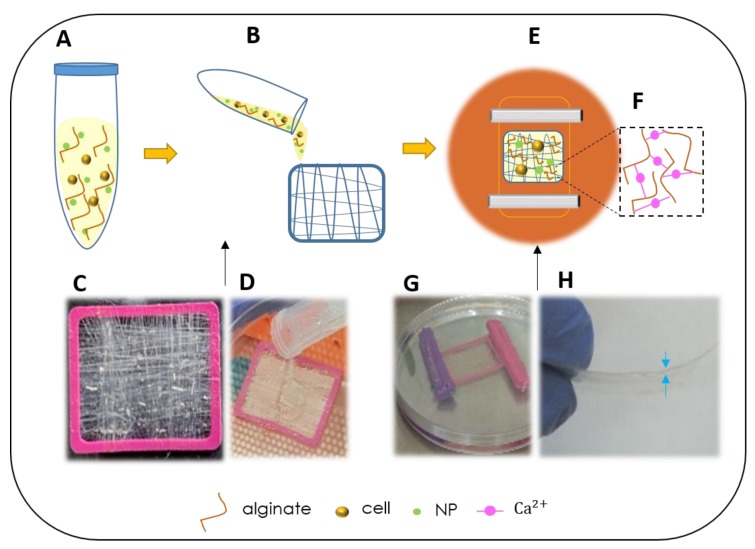
Biocomposite preparation. Alginate solution was mixed with cells or NPs (**A**) or combined with collagen fibers (**B**). The collagen fibers are arranged around a PLA frame to create a multidirectional net (**C**,**D**). The frame is then introduced into a dialysis membrane and submerged in a calcium solution to form the alginate hydrogel (**E**–**G**). The film of biocomposite is ready to use (**H**).

**Figure 2 biomolecules-10-00458-f002:**
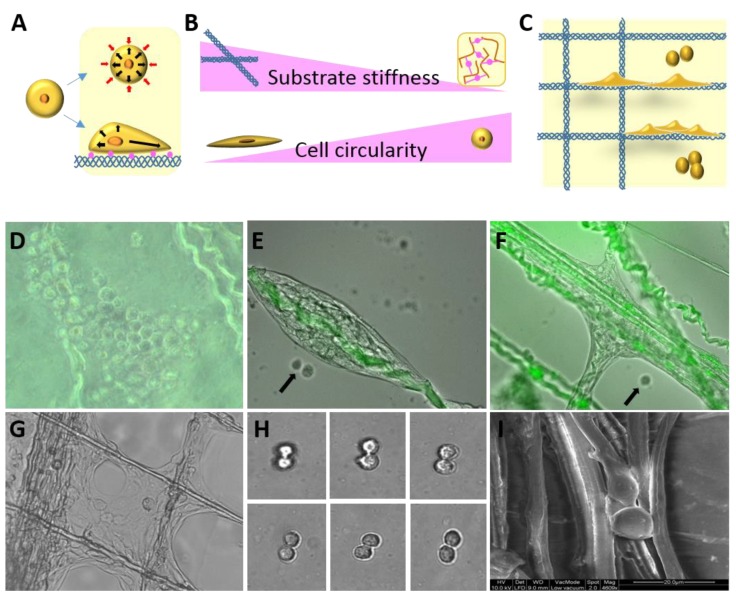
The effect of microenvironment stiffness on cell morphology and growth: scheme showing the morphological effect of differential mechanical forces affecting cells growing on a 2D high-stiffness surface or in a 3D low-stiffness matrix (**A**,**B**). Cell morphology of cells cultured in a collagen-alginate biocomposite (**C**). Light microscopy of cells cultured in a biocomposite, growing in the alginate before reaching a fiber (magnification ×200, **D**). Light and/or fluorescent microscopy of cells forming tissue like structures while cultured in the collagen fiber (green) embedded in alginate biocomposite, (magnification ×200 and 400, **E**–**H**). Phase contrast picture of dividing cells in the alginate (magnification ×400, **H**) and SEM image of dividing cells on the collagen fibers (magnification ×4600, **I**).

**Figure 3 biomolecules-10-00458-f003:**
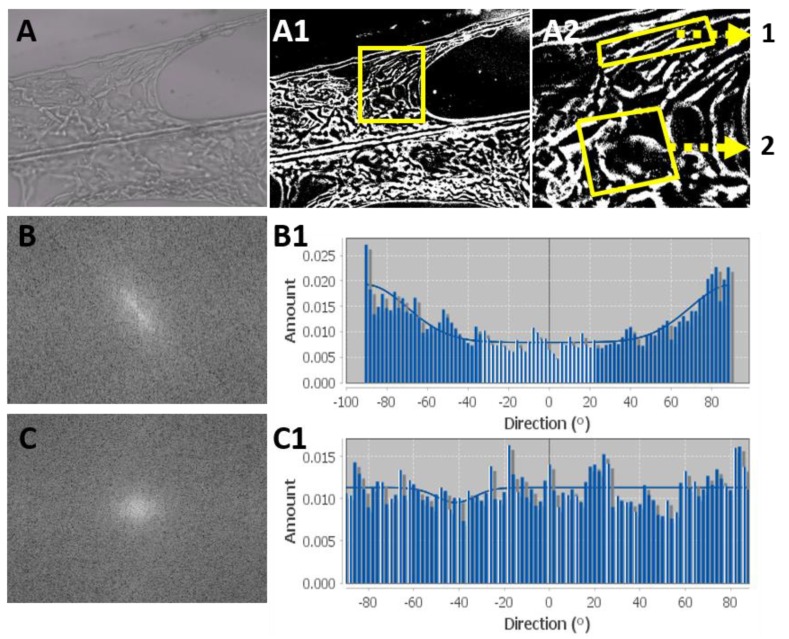
Cellular directional arrangement resembling tissue-like structures in biocomposites: cell orientation was analyzed using binary analysis of the light microscopy images (magnification ×200, **A**). (**A1**) transformation of the image to binary. (**A2**) Magnification of the area marked in A1, showing two different ranges of interest (ROIs): 1, cells next to a fiber; 2, cells growing in tissue like structure between fibers. The ROIs were processed with a fast Fourier transform (FFT) filter and analyzed for directionality. FFT resulting image and histogram of the directionality analysis for ROI1 (**B** and **B1**) and ROI2 (**C** and **C1**).

**Figure 4 biomolecules-10-00458-f004:**
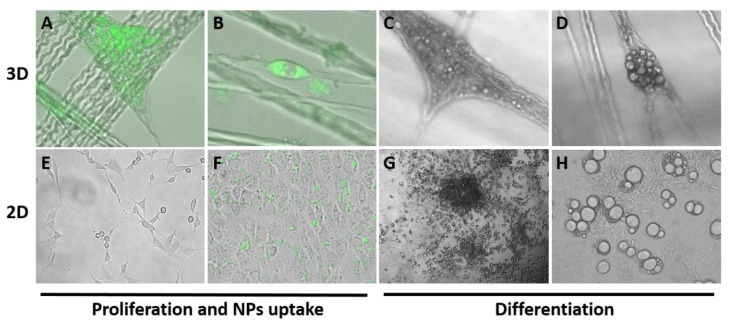
NPs uptake and adipogenic differentiation of cells cultured in collagen-alginate biocomposite. Cells cultured in 3D biocomposite labeled by green fluorescence after Si NPs-FITC conjugated uptake (magnification ×200 and zoom out ×400, **A**,**B**). Adipogenic differentiation of cells cultured on the biocomposite (**C**,**D**). Cells migrated through the biocomposite to reach the culture dish (2D). These cells grew well and were able to take up NPs (magnification ×200 and ×400, **E**,**F**) and continue with adipogenic differentiation (magnification ×40 and ×200, **G**,**H**).

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
