# Peer review of "Mesenchymal Cell Growth and Differentiation on a New Biocomposite Material: A Promising Model for Regeneration Therapy"

_biomolecules, 2020, doi:10.3390/biom10030458_

Round 1

Reviewer 1 Report

Dear Authors

In line 233, You placed  Radiographic Examination

I didn't see  any rx in the paper. I need to observe the radiographic changes  at 4, 8 and 12 weeks in all the implants used.

How you calculated the MBH and Marginal bone loss?

The paper have a lot of empty spaces page 3 to 4, page6 to 7 , 7 to 8, 9 to 10, 11 to 12. Why you leave so many spaces in between the titles?

Please correct

The paper looks great

Author Response

Review for biomolecules-725123

Please find enclose our revised manuscript. We appreciate the evaluation by experts and here are enclosed our answers to the review suggested that were also incorporated in the text and are submitted in the revised paper. The reviewer comments were constructive and improved the revised manuscript and we thank the reviewer for his effort and comments.

Replay to Reviewer 1

According to reviewer comment the manuscript went through English editing and the correction were embedded in the manuscript.

As suggested the method section was added with details.

Specific comment: Just to clarify there were no radiographic in this study

Reviewer 2 Report

Dear authors

The manuscript entitled Mesenchymal cells growth and differentiation on a new Bio-composite material a promise model towards therapy” aim to describe studies showing the biocompatibility of a new Bio-composite material on some aspects of the biology of MSCs. In the introduction the authors illustrate the characteristics and the properties of bio-materials and in particular of the biocomposite and the possibility of their use as new scaffolds.  The topic is very intruing in consideration of the specific characteristics of the proposed bio-composite material. Moreover, albeit the manuscript is well conceived as idea and specific goal to reach is interesting, the organization of materials and methods as well as the presentation of results are not quite accurate and detailed. The authors should describe better the tests used to analyze MSCs proliferation and differentiation in adipocytes. Figure 2, Figure 3 and figure 4 illustrate morphology of cells and proliferation and differentiation.

The authors do not specify the magnification of the microscope for the photo. Moreover, the authors do not describe the markers used to assay the efficiency of the differentiation process of MSCs. Statistic methods to evaluate the efficiency of bio-composite material on the biology of MSCs are not described. On what number of replicates the assays were performed?

In addition, the meaning of several sentences in the introduction and in the discussion are not completely clear and understandable.   

Author Response

Review for biomolecules-725123

Please find enclose our revised manuscript. We appreciate the evaluation by experts and here are enclosed our answers to the review suggested that were also incorporated in the text and are submitted in the revised paper. The reviewer comments were constructive and improved the revised manuscript and we thank the reviewer for his effort and comments.

Replay to Reviewer 2

We thank the reviewer for his comment that the manuscript is well conceived as idea and specific goals to reach is interesting. We corrected the materials and methods sections as well added details in the results.

The MSCs proliferation and differentiation in based on morphological features of the cells that changed their phenotype from fibroblast elongated cells to differentiated adipocyte (they become round cells that accumulated the lipid droplets). We recently published series of studies (listed below) that followed these cell morphology features and are well accepted as gold standard to follow the adipocytes differentiation. The aim of this study is based on the morphological alterations to prove that the mesenchymal cells were physiologically active, viable, proliferated and retained the capacity to differentiate into adipocytes when are grown on the new scaffold presented in this study. This is a crucial and important step that for introducing new material that will satisfying meet the IS0 10993-5 standard for cytotoxicity in vitro towards the development of scaffolding material that support cell normal activity.

We added the requested information

  • Magnification of the micrographs was added in the legend to pictures.
  • The differentiation of cells is based on morphological studies published by us (listed below).
  • Cell staining grown in the bio composite is not possible and all analysis relies on morphological alteration that were analyzed as described in our previous publications (listed below)
  • The assays in this study were aimed to show by morphology cell division, proliferation and formation of tissue-like structures. All these morphological features related to cell physiology were analyzed by various imaging methods employed live imaging or on fixed cells observed by phase contrast and fluorescence microscope (Fig 2-4) and higher resolution of dividing cells by SEM (Fig 2). All assays are quality control of the biomaterial effect on the cells that are grown is the bio-compatible for safety issue and no toxicity on the cells growth and differentiation
  • Series of assays were performed for various aspects presented were repeated in 6 to 8 experiments
  • As advised, the manuscript was English edited.

References

  • Lustig M ….and Benayahu D, Adipogenesis and lipid production in adipocytes …… a living cell-scale model system of diabesity. Biomech. Model. Mechanobiol., 17, 903–913, 2018

  • Mor-Yossef Moldovan L… et al. Benayahu D, Cell shape alteration during adipogenesis is associated with coordinated matrix cues J. Cell. Physiol., 234(4): 3850–3863. 2019.

  • Lustig, M…. et al and D. Benayahu, Noninvasive Continuous Monitoring of Adipocyte Differentiation: From Macro to Micro Scales. Microsc. Microanal. 25 (1), 119–128, 2019.

Round 2

Reviewer 1 Report

Dear Authors

Can you please explain the Title?

Mesenchymal cell growth and differentiation on a

new. 

This is the only question

The paper is well conducted and answered all questions

Author Response

Please see the file in attachment.

Reviewer 2 Report

the authors modified the text according to the suggestions of the reviewer.

In particular, the chapter of materials and methods was significatively improved by adding more information to the readers. In addition, results and discussions were lightly modified higlighting some interesting aspects. The manuscript now is suitable for publication improved.

Author Response

Please see the file in attachment.
